# Evaluation of first and second trimester maternal thyroid profile on the prediction of gestational diabetes mellitus and post load glycemia

Daniela Mennickent[1,2,3], Bernel Ortega-Contreras[1], Sebastián Gutiérrez-Vega[1], Erica Castro[4], Andrés Rodríguez[3,5,6], Juan Araya[2,3]*, Enrique Guzmán-Gutiérrez[1,3,6]*

1 Facultad de Farmacia, Departamento de Bioquímica Clínica e Inmunología, Universidad de Concepción, Concepción, Chile, 2 Facultad de Farmacia, Departamento de Análisis Instrumental, Universidad de Concepción, Concepción, Chile, 3 Machine Learning Applied in Biomedicine (MLAB), Concepción, Chile, 4 Facultad de Ciencias de la Salud, Departamento de Obstetricia y Puericultura, Universidad de Atacama, Copiapó, Chile, 5 Facultad de Ciencias, Departamento de Ciencias Básicas, Universidad del Bío-Bío, Chillán, Chile, 6 Group of Research and Innovation in Vascular Health (GRIVAS-Health), Chillan, Chile

* eguzman@udec.cl (EG-G); jarayaq@udec.cl (JA)

**Data Availability Statement:** All relevant data are within the manuscript and its Supporting Information files.

## Abstract

Maternal thyroid alterations have been widely associated with the risk of gestational diabetes mellitus (GDM). This study aims to 1) test the first and the second trimester full maternal thyroid profile on the prediction of GDM, both alone and combined with non-thyroid data; and 2) make that prediction independent of the diagnostic criteria, by evaluating the effectiveness of the different maternal variables on the prediction of oral glucose tolerance test (OGTT) post load glycemia. Pregnant women were recruited in Concepción, Chile. GDM diagnosis was performed at 24–28 weeks of pregnancy by an OGTT (n = 54 for normal glucose tolerance, n = 12 for GDM). 75 maternal thyroid and non-thyroid parameters were recorded in the first and the second trimester of pregnancy. Various combinations of variables were assessed for GDM and post load glycemia prediction through different classification and regression machine learning techniques. The best predictive models were simplified by variable selection. Every model was subjected to leave-one-out cross-validation. Our results indicate that thyroid markers are useful for the prediction of GDM and post load glycemia, especially at the second trimester of pregnancy. Thus, they could be used as an alternative screening tool for GDM, independently of the diagnostic criteria used. The final classification models predict GDM with cross-validation areas under the receiver operating characteristic curve of 0.867 (p<0.001) and 0.920 (p<0.001) in the first and the second trimester of pregnancy, respectively. The final regression models predict post load glycemia with cross-validation Spearman r correlation coefficients of 0.259 (p = 0.036) and 0.457 (p<0.001) in the first and the second trimester of pregnancy, respectively. This investigation constitutes the first attempt to test the performance of the whole maternal thyroid profile on GDM and OGTT post load glycemia prediction. Future external validation studies are needed to confirm these findings in larger cohorts and different populations.

**Funding:** This work was financed by National Agency for Research and Development (ANID, https://www.anid.cl/), University of Concepción (UdeC, https://www.udec.cl/), Ministry of Education (MINEDUC, https://www.mineduc.cl/) and Universidad del Bio-Bío (UBB, https://www.ubiobio.cl/). DM was supported by ANID-PhD scholarship 21190736 and UdeC/MINEDUC-Project UCO 1866 fund; BO-C by ANID-MSc scholarship 22201750; AR by UBB 180709 3/R DIUBB grant, and ANID-FONIS SA2210028; JA by ANID-FONDECYT 11181153; and EG-G by ANID-FOVI210057, and ANID-FONDECYT 11170710. The funders had no role in study design, data collection and analysis, decision to publish, or preparation of the manuscript. There was no additional external funding received for this study.

**Competing interests:** The authors have declared that no competing interests exist.

## 1. Introduction

Gestational diabetes mellitus (GDM) is a hyperglycemia state that is diagnosed at the second or third trimester of pregnancy, with glycemia values that vary depending on the diagnostic criteria [1]. This disorder has been linked to several negative effects on maternal and fetal health, short and long-term [2–4]. The improvement of its diagnosis, whether with early detection tools or with alternative screening tools, could help to prevent those adverse pregnancy outcomes. Predictive models, with "prediction" understood as the forecast of future behaviors or unobserved outcomes [5], are an auspicious means to achieve that goal.

GDM prediction has been typically addressed with strategies based on classic clinical risk factors for this pregnancy complication, i.e. maternal age, body mass index (BMI), family history of diabetes, previous GDM, among others. Nevertheless, approaches that consider only in this type of parameters usually have a poor predictive performance for GDM [6–8]. Thus, other biomarkers should be investigated and tested in order to complement and boost them.

Maternal thyroid alterations in the first and the second trimester of pregnancy have been widely related to the risk of GDM [9–12]. Moreover, certain authors have assessed first and second trimester thyroid parameters on the prediction of this pregnancy disorder [13–19]. However, the majority of them considered only a few thyroid markers, instead of the whole thyroid profile [13–18]. Furthermore, various of them didn't evaluate these markers in association with other variables [13–15], which is usually related to an improved predictive accuracy [20]. In addition, even though some authors used thyroid data in multivariate analysis, they didn't assess the particular effect of the thyroid profile on GDM prediction, whether by testing it solo or by adding/removing it from the analysis [16–19].

Therefore, this investigation aims to: 1) evaluate the first and the second trimester maternal thyroid profile on GDM prediction, both alone and combined with non-thyroid parameters; and 2) make that prediction independent of the diagnostic criteria, by testing the performance of the different maternal variables on post load glycemia prediction.

## 2. Materials and methods

### 2.1. Ethical aspects

This biomedical study was approved by the Ethics Committee of the Servicio de Salud Concepción (17-12-88) and was performed in accordance to the World Medical Association Declaration of Helsinki.

### 2.2. Subjects recruitment

Pregnant women with 10–14 gestational weeks were recruited between 2017 and 2019 at three primary health centers (CESFAM Victor Manuel Fernández, CESFAM Santa Sabina and CESFAM Tucapel) in Concepción, Chile. Subjects with pregestational diabetes mellitus, pregestational thyroid disorders, or any pregnancy pathologies different than GDM were excluded. Women that gave their written informed consent were incorporated into the study, and followed through pregnancy until 24–28 gestational weeks.

### 2.3. Blood samples collection

Blood samples were collected on the first (at 10–14 gestational weeks) and the second (at 24–28 gestational weeks) trimester of pregnancy, either after 12 hours of fasting or after 2 hours of a 75 g glucose load. Samples were transported to laboratory at 4°C. Sera (obtained from tubes with no anticoagulant) and plasma (obtained from tubes with sodium fluoride/citrate as

anticoagulant) were aliquoted and stored at -80˚C. White blood cells, obtained from EDTA tubes, were stored at -20˚C.

## 2.4. GDM diagnosis and study groups

Pregnant women were subjected to an oral glucose tolerance test (OGTT) at 24–28 gestational weeks. Subjects with post load glycemia > 140 mg/dL (75 g, 2 h) were diagnosed with GDM. In this study, out of 66 subjects, 12 had GDM. The remaining 54 exhibited no alteration in fasting or post load glycemia, i.e. had normal glucose tolerance (NGT).

## 2.5. Clinical data collection

First trimester maternal clinical variables were obtained from both health centers records and pregnant women statements. Institution-derived parameters involved: age, height, weight, BMI, systolic and diastolic pressure at the first trimester of pregnancy. Self-reported data comprised: supplement consumption, hyperemesis, vaginal bleeding and drug use at the first trimester of pregnancy; and preconception information, i.e. drug use before pregnancy, prior pregnancy complications, prior non-viable pregnancy, fertility problems, polycystic ovary syndrome (PCOS) history, first period age, last period date, personal and family morbid history.

## 2.6. Biochemical determinations

Fasting and post load plasma glucose were measured using the hexokinase method [21]. Thyroid stimulating hormone (TSH), total triiodothyronine (TT3), total thyroxine (TT4), free T4 (FT4), thyroglobulin (TG), TG antibody (aTG), thyroid peroxidase antibody (aTPO), TSH receptor antibody (TRAb) and fat mass and obesity-associated (FTO) rs9939609 genotype were determined as described by Araya et al [22]. Briefly, serum thyroid hormones and antibodies were measured by different immunoassays. For FTO genotyping, genomic DNA was isolated from white blood cells, and then analyzed by a previously optimized PCR-HRM method. Melt curve results were normalized to identify the subject's genotype by allele comparison with control samples of known genotype.

## 2.7. Univariate data analysis

Between the first and second trimester of pregnancy, a total of 75 maternal clinical and biochemical variables were recorded (S1 Dataset). Qualitative data were compared by two-sided Fisher exact test. Quantitative data normality was assessed through Shapiro-Wilk test. Normally distributed variables were compared by unpaired Student t test. Not-normally distributed variables were compared by Mann-Whitney or Wilcoxon matched-pairs signed rank test in unpaired or paired analyses, respectively. Values of p<0.05 were considered statistically significant. Univariate analyses were performed with the software GraphPad Prism 9.2.0 (GraphPad Software Inc, USA).

## 2.8. Multivariate data analysis

For multivariate analyses, almost all the registered maternal variables were used as predictors, except for height, weight and OGTT glycemia. Height and weight weren't considered as individual predictors. Instead, the BMI, which comprises both, was utilized. The OGTT glycemia was employed as a label (i.e. NGT or GDM, for exploration) or as the property to predict (i.e. NGT or GDM, for classification; or the OGTT post load glycemia value, for regression).

For classification and regression analyses, the predictors were divided into four categories: thyroid from the first (Thy1T) and the second (Thy2T) trimester of pregnancy, which

comprised TSH, TT3, TT4, FT4, TG, aTG, aTPO and TRAb; and non-thyroid from the first (NoThy1T) and the second (NoThy2T) trimester of pregnancy, which included every other recorded predictor. All the possible combinations of these four categories were assessed for GDM and post load glycemia prediction.

Prior to multivariate analyses, qualitative variables were transformed into categorical and all data were normalized by auto-scaling.

**2.8.1. Exploratory analysis.** Full auto-scaled data were explored by principal components analysis (PCA). To detect possible outliers, four parameters were evaluated at this stage: sample residual, F ratio, probability and Mahalanobis distance. Subjects were considered outliers if two or more parameters suggested such behavior. Data exploration was performed with the software Pirouette 4.5 (Infometrix Inc, USA).

**2.8.2. Classification analysis.** Different subgroups of auto-scaled data were assessed for GDM prediction through five classification ML techniques: logistic regression (LR), linear support vector machine (L-SVM), partial least squares discriminant analysis (PLS-DA), classification and regression tree (CART) and extreme gradient boosting (XGB). Every model was subjected to leave-one-out cross-validation. LR, PLS-DA and XGBoost were executed in PLS_Toolbox R8.9.2 (Eigenvector Research Inc, USA). L-SVM and CART were implemented by coding in MATLAB R2021a (The MathWorks Inc, USA).

The classification predictive performance was assessed by the determination of the models sensitivity (Se), specificity (Sp) and non-error rate (NER) in both calibration and cross-validation. These parameters were calculated as follows:

$$Se\ (\%) = \frac{TP}{TP + FN} \cdot 100 \tag{1}$$

$$Sp\ (\%) = \frac{TN}{TN + FP} \cdot 100 \tag{2}$$

$$NER\ (\%) = \frac{Se + Sp}{2} \tag{3}$$

Where TP, FN, TN and FP are the number of true positives, false negatives, true negatives and false positives, respectively.

The area under the receiver operating characteristic curve (AUC) of the final classification models was determined through the software GraphPad Prism 9.2.0 (GraphPad Software Inc, USA).

**2.8.3. Regression analysis.** Different subgroups of auto-scaled data were assessed for post load glycemia prediction through the regression ML technique partial least squares (PLS). Every model was subjected to leave-one-out cross-validation. PLS was executed in PLS_Toolbox R8.9.2 (Eigenvector Research Inc, USA).

The regression predictive performance was assessed by the determination of the models root mean square error (RMSE) and relative error (RE) in both calibration (RMSEC and REC) and cross validation (RMSECV and RECV). These parameters were calculated as follows:

$$RMSEC = \sqrt{\frac{\sum_{i=1}^{n} (\hat{y}_i - y_i)^2}{n - 1}} \tag{4}$$

$$RMSECV = \sqrt{\frac{\sum_{i=1}^{n} (\hat{y}_i - y_i)^2}{n}} \tag{5}$$

$$REC\ (\%) = \frac{RMSEC}{\bar{y}} \cdot 100 \tag{6}$$

$$RECV\ (\%) = \frac{RMSECV}{\bar{y}} \cdot 100 \tag{7}$$

Where $\hat{y}_i$ is the predicted post load glycemia for subject i, $y_i$ is the actual post load glycemia for subject i, n is the number of subjects in calibration and cross-validation, and $\bar{y}$ is the mean of the actual post load glycemia values in calibration.

The Spearman r correlation coefficient of the final regression models was determined by the software GraphPad Prism 9.2.0 (GraphPad Software Inc, USA).

**2.8.4. Variable selection.**   Variable importance was retrieved from PLS-DA and PLS regression vectors. For every model, the contribution of each variable was calculated as follows:

$$Variable\ contribution\ (\%) = \frac{|b_i|}{\sum_{i=1}^{n} |b_i|} \cdot 100 \tag{8}$$

Where $b_i$ is the regression vector value for variable i, and n is the number of variables in the classification or regression model.

For each model, variables were sorted according to their percentage contribution. Toward model simplification, the j variables with the highest percentage contribution were selected, with j ranging from 10 to 1.

## 3. Results and discussion

### 3.1. Description of the study groups

To characterize and compare the study groups, classic univariate techniques were used. Table 1 shows only the first and second trimester maternal variables that differed significantly between NGT and GDM pregnancies in the univariate approach. The full list, which contains the 75 maternal variables that were recorded and analyzed in this study, is presented in S1 Table.

In the first trimester of pregnancy, the BMI and the systolic pressure were higher in the GDM group, than in the NGT group. Moreover, prior history of GDM, family history of type 2 diabetes (DM2) and family history of hypertension were more frequent in GDM pregnancies. These observations are consistent with a recent Chilean epidemiologic study, in which subjects with a higher pre-pregnancy BMI, previous record of hypertension or GDM, or family history of DM2 or hypertension, presented a greater risk of GDM [23]. Likewise, personal history of anemia was more frequent in GDM women. The relationship between anemia and GDM appears to be dependent on the underlying cause of the former. Indeed, whereas iron deficiency decreases the risk of GDM [24], B12 deficiency increases it [25]. Interestingly, although iron deficiency was the most common cause of anemia among Chilean women between 1981 and 2010 [26], the 2010–2011 National Food Consumption Survey revealed that, while approximately 30% of women had a deficient consumption of iron, nearly 60% of them had insufficient intake of vitamin B12 [27]. The FTO genotype was significantly different between the two groups as well: while the TT genotype was more common in NGT women, the TA and the AA genotypes were more frequent in GDM women. Contradictory results have been reported regarding to the association between FTO rs9939609 polymorphism and GDM [28–30]. However, in Chilean adults the risk allele A was linked to an increased BMI [31], which is a risk factor for GDM in this population [23].

**Table 1. Maternal variables that differ significantly between normal glucose tolerance and gestational diabetes mellitus pregnancies in univariate analysis.**

| Variable | | Unit | NGT (n = 54) | GDM (n = 12) | p value | | All (n = 66) |
|---|---|---|---|---|---|---|---|
| **First trimester** | | | | | | | |
| BMI | | Kg/m$^2$ | 27.4 (23.5–31.1) | 30.3 (27.1–31.6) | 0.046 | * | 27.8 (24.5–31.2) |
| Systolic pressure | | mmHg | 109 ± 11 | 116 ± 7 | 0.042 | * | 110 ± 11 |
| Preconception data | | | | | | | |
| Prior GDM | | % | 1.9 (1/54) | 33.3 (4/12) | 0.003 | ** | 7.6 (5/66) |
| Personal history of anemia | | % | 0.0 (0/54) | 16.7 (2/12) | 0.031 | * | 3.0 (2/66) |
| Family history of DM2 | | % | 20.4 (11/54) | 66.7 (8/12) | 0.003 | ** | 28.8 (19/66) |
| Family history of hypertension | | % | 33.3 (18/54) | 66.7 (8/12) | 0.049 | * | 39.4 (26/66) |
| FTO genotype (rs9939609) | TT | % | 53.7 (29/54) | 25.0 (3/12) | 0.047 | * | 48.5 (32/66) |
| | TA | % | 44.4 (24/54) | 66.7 (8/12) | | | 48.5 (32/66) |
| | AA | % | 1.9 (1/54) | 8.3 (1/12) | | | 3.0 (2/66) |
| **Second trimester** | | | | | | | |
| TSH | | μIU/mL | 1.42 (1.10–2.09) | 3.25 (2.28–4.38) | 0.001 | ** | 1.54 (1.10–2.53) |
| TT3 | | ng/mL | 1.92 (1.90–2.04) | 2.02 (1.92–2.37) | 0.027 | * | 1.92 (1.90–2.11) |
| FT4 | | ng/dL | 0.79 (0.76–0.81) | 0.68 (0.63–0.78) | 0.006 | ** | 0.77 (0.76–0.81) |
| aTG | | IU/mL | 4.50 (4.30–7.05) | 16.93 (8.42–18.45) | 0.020 | * | 4.89 (4.31–16.85) |
| Fasting glycemia | | mg/dL | 79 ± 8 | 86 ± 10 | 0.013 | * | 80 ± 9 |
| OGTT glycemia (75 g, 2 h) | | mg/dL | 102 (94–109) | 150 (141–172) | <0.001 | **** | 105 (96–134) |

NGT: Normal glucose tolerance. GDM: Gestational diabetes mellitus. BMI: Body mass index. DM2: Type 2 diabetes. FTO: Fat mass and obesity-associated gene. TSH: Thyroid stimulating hormone. TT3: Total triiodothyronine. FT4: Free thyroxine. aTG: Thyroglobulin antibody. OGTT: Oral glucose tolerance test. Qualitative variables are presented as percentage (proportion); quantitative variables with normal distribution, as mean ± standard deviation; and quantitative variables with non-normal distribution, as median (interquartile range).

* $p<0.05$.

** $p<0.01$.

**** $p<0.0001$.

In the second trimester of pregnancy, TSH, TT3 and aTG were higher in the GDM group, when compared to the NGT group. In addition, FT4 was lower in GDM pregnancies. Preceding studies describe similar results. For instance, Gutierrez-Vega et al reported that GDM subjects had greater levels of TSH and TT3 than NGT subjects at 24–28 weeks of pregnancy [32]. Likewise, Tang et al found increased levels of aTG and decreased levels of FT4 in GDM pregnant women at 17.0 ± 4.0 weeks of gestation [14]. Furthermore, second trimester fasting glycemia and OGTT post load glycemia were higher in the GDM group, which is in agreement with its diagnostic criteria. The prevalence of GDM in this study was 18.2%, moderately superior than the 13.0% described for Chile in 2015 [23] and to the 11.2% reported for South and Central America in 2015–2018 [1].

To better understand the differences observed between NGT and GDM groups in terms of thyroid markers, a complementary analysis was performed. For each group, TSH, TT3, FT4 and aTG levels were compared between the first and the second trimester of pregnancy. The results are displayed in S2 Table. TSH didn't vary significantly in any of the groups, although it tended to increase from the first to the second trimester, especially in GDM subjects. The latter accounts for the statistical difference between NGT and GDM pregnancies regarding TSH2T (Table 1). TT3 raised and FT4 decreased from the first to the second trimester in both groups. The magnitude of those changes was greater in GDM women, which explains the significant difference amidst the two groups in terms of TT32T and FT42T (Table 1). aTG lowered from the first to the second trimester in the NGT subjects, whereas it didn't vary in GDM subjects.

This contrasting behavior gives a rational for the statistical difference between NGT and GDM pregnancies in relation to aTG2T (Table 1). The changes observed in thyroid hormones within the NGT group are consistent with the physiology of normal pregnancy, which is characterized by is a slight rise in TSH, a considerable increase in TT3 and a substantial decrease in FT4 between early- and mid-gestation [33]. The fact that these changes are exacerbated in GDM is not yet fully understood, however it has been proposed that it may be linked to maternal weight. High maternal weight, as high maternal BMI, is an important risk factor for GDM. Haddow et al proposed that higher maternal weight would induce higher peripheral deiodinase activity, which in turn would lead to reduced FT4 and increased TT3. Moreover, higher TT3 would induce higher plasma glucose levels, thereby contributing to GDM pathophysiology [34]. This hypothesis, although just beginning to be tested by other authors [35–37], provides a feasible explanation for the differences observed between GDM and NGT pregnancies in this study, especially considering that the GDM group presented a higher BMI than the NGT group. In a similar manner, the changes observed in aTG within the NGT group are in line with previous reports, in which the levels of this thyroid antibody decrease from early- to mid-gestation in normal pregnancies [38–40]. This decline is due to the physiological immunosuppression that occurs in normal pregnancy to tolerate the fetus and placenta [41]. That process is disrupted in GDM [42], which could explain why the levels of aTG do not decrease amidst trimesters in the GDM group as they do in the NGT group.

These results indicate that, when considered individually, the main differential variables between the two study groups are non-thyroid in the first trimester of pregnancy, and thyroid in the second trimester of pregnancy.

### 3.2. Exploration of maternal data by PCA

To assess the potential of a multivariate approach to distinguish GDM from NGT pregnancies, a PCA was performed. The unsupervised analysis comprised 72 maternal variables and all the 66 subjects. No outliers were found. The PCA results are displayed in Fig 1.

Fig 1A shows that NGT and GDM women are partially separated by the conjunction of principal components 1 (PC1) and 2 (PC2). Fig 1B highlights the maternal variables that, when combined, contribute the most to the separation between the two study groups. They are mainly non-thyroid variables from the first trimester, and thyroid variables from the second trimester of pregnancy, which is consistent with the univariate analysis results. However, although more discreetly, some thyroid variables from the first trimester also appear to contribute to the separation between NGT and GDM subjects. A previous study reported similar results. Araya et al used PCA to explore clinical and biochemical maternal data from NGT and GDM pregnancies. After removing the OGTT post load glycemia from the analysis, the separation between two groups was mostly attributable to first trimester clinical variables and second trimester thyroid parameters, with a minor but noticeable contribution of first trimester thyroid data [22]. These observations are also in agreement with other authors, who described that different second trimester thyroid markers are more strongly associated with the risk of GDM, than their first trimester counterparts [14,16,43–45].

These results suggest that a supervised multivariate model, i.e. a ML-based model, built with first and second trimester thyroid parameters could be successful in the prediction of GDM.

### 3.3. Prediction of GDM with ML techniques

To evaluate the ability of maternal thyroid markers to predict GDM, different classification models were developed, covering various subgroups of predictors and ML techniques. Table 2

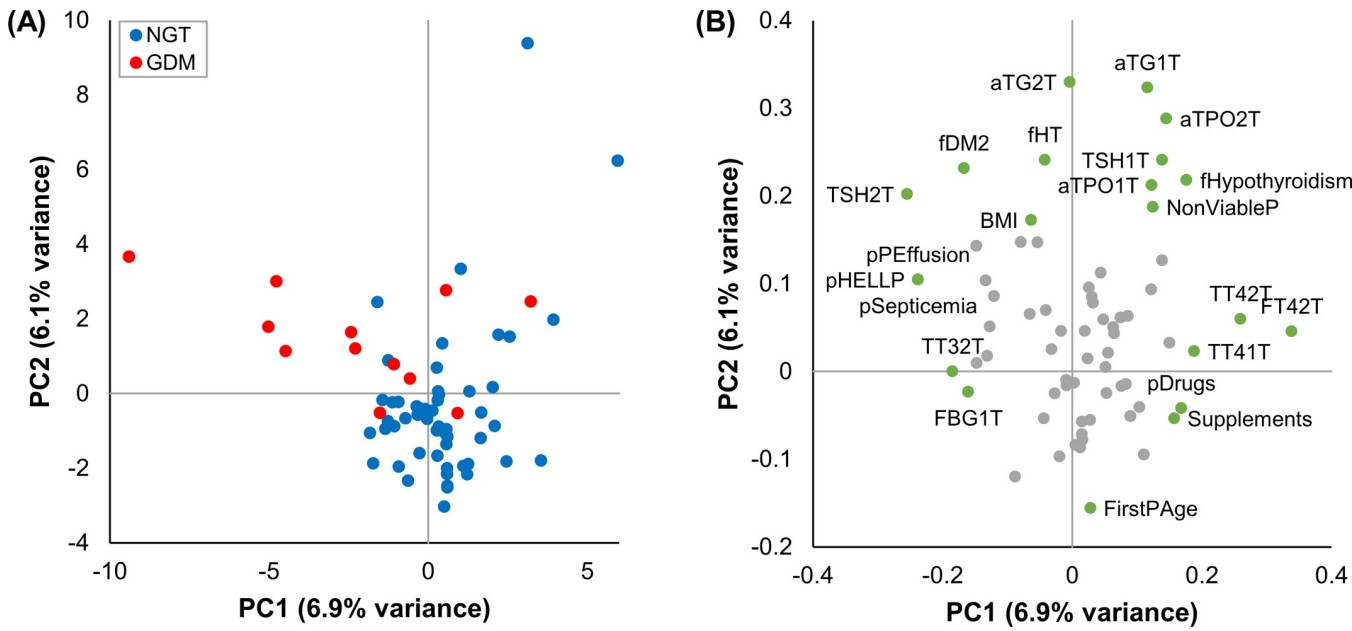

**Fig 1. Exploration of maternal data by principal component analysis.** (A) The scores plot shows the NGT (blue) and GDM (red) subjects projected in the space constituted by PC1 and PC2. (B) The loadings plot shows the variables contribution to PC1 and PC2. The ones that contribute the most to each PC are highlighted (green) and labeled. *NGT*: *Normal glucose tolerance*. *GDM*: *Gestational diabetes mellitus*. *PC*: *Principal component*. *1T*: *First trimester*. *2T*: *Second trimester*. *pHELLP*: *Prior hemolysis elevated liver enzymes and low platelets syndrome*. *pPEffusion*: *Prior pleural effusion*. *pSepticemia*: *Prior septicemia*. *fDM2*: *Family history of type 2 diabetes*. *fHT*: *Family history of hypertension*. *BMI*: *Body mass index*. *fHypothyroidism*: *Family history of hypothyroidism*. *NonViableP*: *Prior non-viable pregnancy*. *pDrugs*: *Drug use before pregnancy*. *Supplements*: *Supplement consumption*. *FirstPAge*: *First period age*. *TSH*: *Thyroid stimulating hormone*. *aTG*: *Thyroglobulin antibody*. *aTPO*: *Thyroid peroxidase antibody*. *TT4*: *Total thyroxine*. *FT4*: *Free thyroxine*. *TT3*: *Total triiodothyronine*. *FBG*: *Fasting blood glucose*.

displays the predictive performance of each model in calibration and cross-validation. It focuses on the NER, which is a global measure of classification quality that doesn't depend on the relative sizes of the groups of interest [46]. In other words, the NER isn't biased by the presence of unbalanced groups, which is the case of this study: the NGT and GDM groups comprise 54 and 12 subjects, respectively. The sensitivity and specificity values of each model are presented in S3 Table.

First trimester thyroid variables (Thy1T) solo displayed a poor performance for the prediction of GDM. Their models got an average cross-validation NER (CV-NER) of 51.3%. Furthermore, in most cases the addition of Thy1T to other variables didn't improve the classification quality. However, the model built only on first trimester predictors that had the highest CV-NER comprises Thy1T (PLS-DA, Thy1T + NoThy1T, CV-NER of 81.0%). The model with second trimester predictors that showed the greatest CV-NER also encompasses Thy1T (PLS-DA, NoThy1T + Thy1T + Thy2T, CV-NER of 87.0%). The top 15 most important variables of those models are presented in Fig 2. Fig 2A concerns the one covering first trimester data only, and Fig 2B the one containing second trimester data. Notably, the 10 major predictors of both models include first trimester thyroid markers, which didn't come up as relevant in univariate analysis. This supports the idea that emerged in exploratory analysis: although moderately, first trimester thyroid variables contribute to the differentiation between NGT and GDM subjects when they are in association with other data. This view is consistent with the study of Zhu et al, who used LR and clinical and biochemical maternal parameters to predict GDM. They observed that the addition of first trimester TT3 and TT3/FT4 improved the AUC of their classification model, from 0.710 to 0.726 and 0.724, respectively [16].

**Table 2. Non-error rate of machine learning models that predict gestational diabetes mellitus based on different maternal variables.**

| Maternal predictors | Calibration NER | | | | | Cross-validation NER | | | | |
|---|---|---|---|---|---|---|---|---|---|---|
| | LR | L-SVM | PLS-DA | CART | XGB | LR | L-SVM | PLS-DA | CART | XGB |
| Thy1T | 57.4 | 54.2 | 62.0 | 50.0 | 75.0 | 51.4 | 50.0 | 52.8 | 50.0 | 52.3 |
| Thy2T | 86.6 | 83.3 | 83.8 | 79.2 | 100.0 | 77.3 | 74.1 | 81.9 | 79.2 | 82.4 |
| Thy1T + Thy2T | 95.8 | 79.2 | 93.1 | 79.2 | 100.0 | 59.3 | 74.1 | 81.0 | 79.2 | 83.3 |
| NoThy1T | 100.0 | 83.3 | 93.1 | 78.2 | 100.0 | 75.0 | 73.1 | 80.1 | 73.1 | 59.7 |
| NoThy1T + Thy1T | 100.0 | 91.7 | 100.0 [a] | 65.7 | 100.0 | 75.9 | 73.1 | 81.0 [a] | 61.6 | 61.6 |
| NoThy1T + Thy2T | 100.0 | 91.7 | 88.9 | 79.2 | 100.0 | 74.1 | 83.3 | 86.1 | 79.2 | 79.2 |
| NoThy1T + Thy1T + Thy2T | 100.0 | 87.5 | 93.1 [b] | 79.2 | 100.0 | 75.0 | 83.3 | 87.0 [b] | 79.2 | 79.2 |
| NoThy2T | 54.2 | 50.0 | 69.0 | 50.0 | 50.0 | 54.2 | 50.0 | 64.8 | 50.0 | 50.0 |
| NoThy2T + Thy1T | 62.5 | 54.2 | 58.3 | 50.0 | 58.3 | 54.6 | 54.2 | 48.1 | 50.0 | 49.1 |
| NoThy2T + Thy2T | 91.7 | 87.5 | 86.1 | 79.2 | 100.0 | 77.3 | 83.3 | 81.0 | 79.2 | 83.3 |
| NoThy2T + Thy1T + Thy2T | 95.8 | 87.5 | 92.1 | 79.2 | 100.0 | 65.7 | 78.2 | 84.3 | 79.2 | 83.3 |
| NoThy1T + NoThy2T | 100.0 | 95.8 | 89.8 | 78.2 | 100.0 | 75.9 | 77.3 | 85.2 | 65.7 | 63.0 |
| NoThy1T + NoThy2T + Thy1T | 100.0 | 91.7 | 94.0 | 65.7 | 100.0 | 73.1 | 80.6 | 79.2 | 61.6 | 60.6 |
| NoThy1T + NoThy2T + Thy2T | 100.0 | 91.7 | 88.9 | 79.2 | 100.0 | 69.0 | 83.3 | 86.1 | 79.2 | 79.2 |
| NoThy1T + NoThy2T + Thy1T + Thy2T | 100.0 | 95.8 | 93.1 | 79.2 | 100.0 | 67.1 | 83.3 | 86.1 | 79.2 | 79.2 |

[a] Model with the highest NER for GDM prediction, using 1T data only.

[b] Model with the highest NER for GDM prediction, including 2T data.

NER: Non-error rate. LR: Logistic regression. L-SVM: Linear support vector machine. PLS-DA: Partial least squares discriminant analysis. CART: Classification and regression tree. XGB: Extreme gradient boosting. Thy: Thyroid predictors. NoThy: Non-thyroid predictors. 1T: First trimester. 2T: Second trimester.

The particular first trimester thyroid variables that are among the top 10 most relevant predictors of the presented models are TG1T, in the first and the second trimester approaches, and TRAb1T, in the first trimester approach. The levels of TG1T and TRAb1T tend to be higher in the GDM group than in the NGT group (S1 Table), yet they didn't reach statistical significance in univariate analysis. Remarkably, in both models they are more important than age and BMI, two known risk factors for GDM [23]. Few articles have studied the association between these two thyroid markers and the risk of GDM. Bell et al assessed the role of TG1T on the risk of GDM in Finnish pregnant women, and found no association between them, neither solo nor combined with blood iodine levels [47]. Likewise, Wang et al evaluated the association between TRAb1T and the risk of GDM in Shanghainese pregnant women, and found that positive TRAb1T was related to a decreased risk of this pregnancy disorder [48]. Pregnant women from both Finland [47] and Shanghai [48] are known to be iodine deficient. Differently from those populations, Chilean pregnant women have an adequate iodine intake [49], which could partially explain these contrasting results. It is also important to mention that, in the present study, TG1T and TRAb1T only came out as relevant for the prediction of GDM in association with other maternal variables, which wasn't comprehensively assessed in the mentioned studies. As far as we know, neither TG1T nor TRAb1T had been tested for the prediction of GDM before.

In contrast to Thy1T, second trimester thyroid variables (Thy2T) solo showed an acceptable performance for GDM prediction. Their models reached an average CV-NER of 79.0%. Moreover, the addition of Thy2T to other variables improved the classification performance in most cases. Interestingly, Thy2T presented a higher CV-NER than second trimester non-thyroid variables (NoThy2T, average CV-NER of 53.8%), and the only NoThy2T assessed in this study was fasting blood glucose (FBG2T). This means that Thy2T had a better predictive performance than FBG2T, a parameter that is often used as a screening tool for GDM [50].

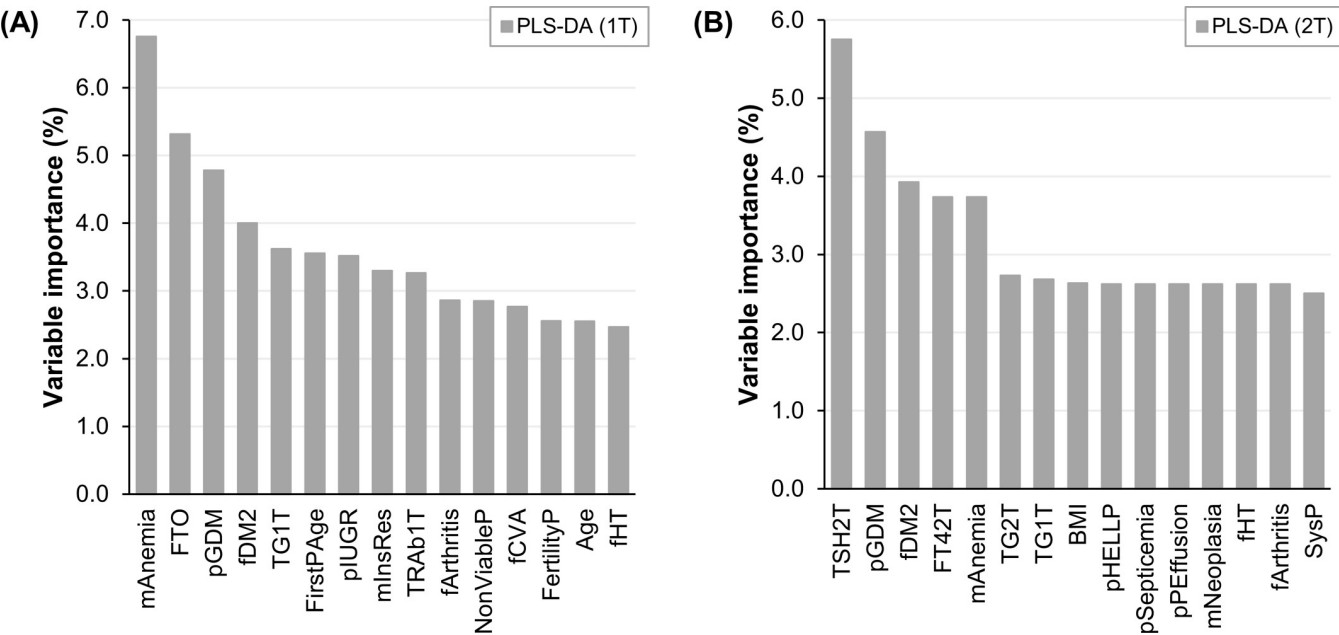

**Fig 2. Variable importance of the machine learning models that predict gestational diabetes mellitus with best performance.** (A) Built on first trimester data only. (B) Including second trimester data. *1T: First trimester. 2T: Second trimester. PLS-DA: Partial least squares discriminant analysis. mAnemia: Personal anemia. FTO: Fat mass and obesity-associated genotype (rs9939609). pGDM: Prior gestational diabetes mellitus. fDM2: Family history of type 2 diabetes. TG: Thyroglobulin. FirstPAge: First period age. pIUGR: Prior intrauterine growth restriction. mInsRes: Personal insulin resistance. TRAb: TSH receptor antibody. fArthritis: Family history of rheumatoid arthritis. NonViableP: Prior non-viable pregnancy. fCVA: Family history of cerebrovascular accident. FertilityP: Fertility problems. fHT: Family history of hypertension. TSH: Thyroid stimulating hormone. FT4: Free thyroxine. BMI: Body mass index. pHELLP: Prior hemolysis elevated liver enzymes and low platelets syndrome. pSepticemia: Prior septicemia. pPEffusion: Prior pleural effusion. mNeoplasia: Personal neoplasia. SysP: Systolic pressure.*

Moreover, several second trimester thyroid parameters figure among the top 15 most important predictors of the second trimester model, whereas FBG2T doesn't. Therefore, Thy2T could be employed as an alternative screening tool for GDM. To our knowledge, this is the first time that thyroid markers are proposed to be used with that purpose.

The particular second trimester thyroid variables that are part of the top 10 most relevant predictors of the second trimester model are TSH2T, FT42T and TG2T. TSH2T and FT42T arose as significant in univariate analysis, but TG2T didn't. They are more important to the model than age and BMI. Furthermore, TSH2T is more relevant than prior history of GDM and family history of DM2, other known risk factors for this pregnancy complication [23]. Higher TSH2T [43,44] and lower FT42T [14,16,45] levels have been associated to an increased risk of GDM, whereas TG2T hadn't been linked with the risk of this disorder until now. The levels of this hormone precursor tend to be higher in the GDM group, than in the control group (S1 Table). FT42T has been used for its prediction in univariate analysis, with poor classification performance [14]. However, as far as we can tell, none of these thyroid markers had been used for the prediction of GDM in multivariate analysis before.

These results prove that maternal thyroid variables are useful for the prediction of GDM, especially at the second trimester of pregnancy.

### 3.4. Prediction of post load glycemia with ML techniques

In order to make the prediction of GDM independent from the diagnostic criteria, the prediction of post load glycemia was conducted. Whereas the prediction of GDM is a binary classification task, the prediction of the continuous parameter post load glycemia is a regression task.

Given that PLS-DA stood out in classification, its related ML technique PLS was used for regression. The prediction performance was evaluated by RMSE and RE (also known as normalized RMSE or coefficient of variation of the RMSE), which are usually employed to assess the error of regression models in ML-based studies [51]. Table 3 presents the RE of each PLS model in calibration and cross-validation. Their RMSEs are shown in S4 Table.

Thy1T and Thy2T exhibited a similar behavior than the observed in the classification analysis. Thy1T alone presented the highest error for post load glycemia prediction (RECV of 24.7%) and their addition to other variables didn't improve the regression performance. Thy2T alone displayed a better performance (RECV of 20.1%) and their incorporation to other predictors enhanced the regression quality. Moreover, Thy2T had a lower RECV than FBG2T (NoThy2T, RECV of 22.4%). This is in agreement with the study of Ouzilleau et al, who found a very weak correlation between FBG and OGTT post load glycemia (75 g, 2 h) at 24–28 weeks of pregnancy [52].

In opposition with the observed in the classification analysis, the regression models with the lowest RECV don't contain Thy1T, neither when built on first trimester predictors only (PLS, NoThy1T, RECV of 21.4%) nor when include second trimester predictors (PLS, NoThy1T + NoThy2T + Thy2T, RECV of 18.7%). However, the latter does include Thy2T. The top 15 most relevant variables of these models are presented in Fig 3. Fig 3A concerns the one comprising first trimester data only, and Fig 3B the one involving second trimester data.

The particular thyroid variables that are among the top 10 most important predictors of the second trimester model are TSH2T, FT42T and TT42T. In line with the results described above, TSH2T and FT42T are more relevant to the model than FBG2T. Moreover, the three thyroid markers are more relevant than family history of DM2 and prior GDM, two recognized risk factors for GDM in Chile [23] and worldwide [1]. In contrast to TSH2T and FT42T,

**Table 3. Relative error of machine learning models that predict post load glycemia based on different maternal variables.**

| Maternal predictors | PLS | |
|---|---|---|
| | Calibration RE (%) | Cross-validation RE (%) |
| Thy1T | 22.3 | 24.7 |
| Thy2T | 16.3 | 20.1 |
| Thy1T + Thy2T | 17.7 | 22.9 |
| NoThy1T | 16.3 [a] | 21.4 [a] |
| NoThy1T + Thy1T | 15.7 | 21.6 |
| NoThy1T + Thy2T | 15.2 | 19.1 |
| NoThy1T + Thy1T + Thy2T | 14.8 | 20.0 |
| NoThy2T | 21.6 | 22.4 |
| NoThy2T + Thy1T | 21.0 | 23.7 |
| NoThy2T + Thy2T | 17.4 | 20.1 |
| NoThy2T + Thy1T + Thy2T | 17.1 | 22.2 |
| NoThy1T + NoThy2T | 15.5 | 20.9 |
| NoThy1T + NoThy2T + Thy1T | 15.0 | 21.2 |
| NoThy1T + NoThy2T + Thy2T | 14.7 [b] | 18.7 [b] |
| NoThy1T + NoThy2T + Thy1T + Thy2T | 14.3 | 19.6 |

[a] Model with the lowest RE for post load glycemia prediction, using 1T data only.

[b] Model with the lowest RE for post load glycemia prediction, including 2T data.

PLS: Partial least squares. RE: Relative error. 1T: First trimester. 2T: Second trimester. Thy: Thyroid predictors. NoThy: Non-thyroid predictors.

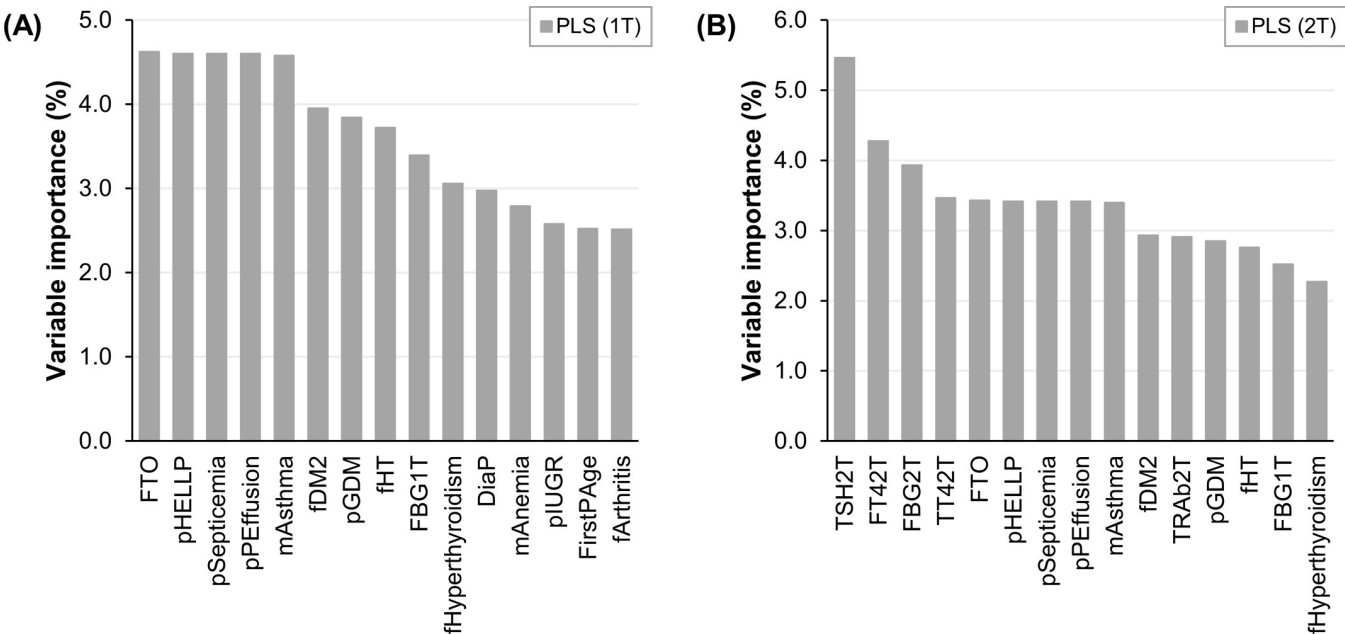

**Fig 3. Variable importance of the machine learning models that predict post load glycemia with best performance.** (A) Built on first trimester data only. (B) Including second trimester data. *1T: First trimester. 2T: Second trimester. PLS: Partial least squares. FTO: Fat mass and obesity-associated genotype (rs9939609). pHELLP: Prior hemolysis elevated liver enzymes and low platelets syndrome. pSepticemia: Prior septicemia. pPEffusion: Prior pleural effusion. mAsthma: Personal asthma. fDM2: Family history of type 2 diabetes. pGDM: Prior gestational diabetes mellitus. fHT: Family history of hypertension. FBG: Fasting blood glucose. fHyperthyroidism: Family history of hyperthyroidism. DiaP: Diastolic pressure. mAnemia: Personal anemia. pIUGR: Prior intrauterine growth restriction. FirstPAge: First period age. fArthritis: Family history of rheumatoid arthritis. TSH: Thyroid stimulating hormone. FT4: Free thyroxine. TT4: Total thyroxine. TRAb: TSH receptor antibody.*

TT42T didn't emerge as significant in univariate analysis and hadn't been linked to the risk of GDM before. TT42T levels tend to be lower in the GDM group, than in the NGT group (S1 Table). To our knowledge, this is the first time that these thyroid parameters are used to predict post load glycemia in pregnancy. In agreement with was already suggested, they could be used as an alternative screening tool for GDM, independently of its diagnostic criteria.

These results demonstrate that second trimester maternal thyroid variables are useful for the prediction of OGTT post load glycemia.

## 3.5. Models simplification by variable selection

In order to ease the application of the best classification and regression models, variable selection was performed. Prior ML-based studies carried out similar approaches [6,19]. For each full model, different numbers of variables were selected, which resulted in various simplified models. Their performance is presented in S5 Table. The first and second trimester simplified models with the highest CV-NER (in the case of the prediction of GDM) and the lowest RECV (in the case of the prediction of post load glycemia) were chosen as the final ones. Table 4 displays the NER or RE of the full and the final simplified models, in calibration and cross-validation.

In classification with first trimester data only, the full model achieved a CV-NER of 81.0%. The selection of the top 4 most important variables increased the CV-NER to 87.0%. In the case of classification including second trimester data, the full model reached a CV-NER of 87.0%. The selection of the top 9 most relevant variables improved the predictive performance, but in a more moderate extent than in the first trimester case (CV-NER of 89.8%). The former

**Table 4. Non-error rate and relative error of machine learning models that predict gestational diabetes mellitus and post load glycemia before and after variable selection.**

| Maternal predictors | Calibration | | Cross-validation | |
|---|---|---|---|---|
| | PLS-DA | PLS | PLS-DA | PLS |
| | NER (%) | RE (%) | NER (%) | RE (%) |
| *First trimester* | | | | |
| Full [a] | 100.0 | 16.3 | 81.0 | 21.4 |
| Simplified [b] | 87.0 | 17.6 | 87.0 | 20.7 |
| *Second trimester* | | | | |
| Full [c] | 93.1 | 14.7 | 87.0 | 18.7 |
| Simplified [d] | 89.8 | 16.1 | 89.8 | 18.4 |

[a] For PLS-DA: NoThy1T + Thy1T. For PLS: NoThy1T.

[b] For PLS-DA: 4 predictors (mAnemia, FTO, pGDM, fDM2). For PLS: 7 predictors (FTO, pHELLP, pSepticemia, pPEffusion, mAsthma, fDM2, pGDM).

[c] For PLS-DA: NoThy1T + Thy1T + Thy2T. For PLS: NoThy1T + NoThy2T + Thy2T.

[d] For PLS-DA: 9 predictors (TSH2T, pGDM, **fDM2**, FT42T, mAnemia, TG2T, TG1T, BMI, pHELLP). For PLS: 9 predictors (TSH2T, FT42T, FBG2T, TT42T, FTO, pHELLP, pSepticemia, pPEffusion, mAsthma).

PLS-DA: Partial least squares discriminant analysis. PLS: Partial least squares. NER: Non-error rate. RE: Relative error. NoThy: Non-thyroid predictors. Thy: Thyroid predictors. 1T: First trimester. 2T: Second trimester. mAnemia: Personal anemia. FTO: Fat mass and obesity-associated genotype (rs9939609). pGDM: Prior gestational diabetes mellitus. fDM2: Family history of type 2 diabetes. pHELLP: Prior hemolysis elevated liver enzymes and low platelets syndrome. pSepticemia: Prior septicemia. pPEffusion: Prior pleural effusion. mAsthma: Personal asthma. TSH: Thyroid stimulating hormone. FT4: Free thyroxine. TG: Thyroglobulin. BMI: Body mass index. FBG: Fasting blood glucose. TT4: Total thyroxine.

simplified model doesn't contain thyroid predictors, but the latter includes both first and second trimester thyroid markers. It is important to notice that the second trimester model has a better predictive performance than the first trimester model, both before and after variable selection. This could be related to the fact that GDM's hyperglycemia state is already established at the second trimester of pregnancy. In contrast, at the first trimester of pregnancy the pathophysiological events that lead to the clinical manifestation of GDM are likely to be incipient. Thus, the phenotypic alterations induced by this pregnancy complication will probably be more evident in the second trimester than in the first trimester of pregnancy, which is reflected in the variables' behavior and the models' performance. This idea is consistent with was described by other authors. For instance, Catalano et al observed an impaired insulin response and a reduced hepatic insulin sensitivity in GDM women at late gestation, but not at early gestation [53]. Likewise, Wong et al found an increased placental volume in GDM subjects at the second trimester, but not at the first trimester of pregnancy [54]. Similarly, Sovio et al noticed an altered fetal biometry in GDM pregnancies at 28 weeks of gestation, but not at 20 weeks of gestation [55]. On the other hand, the main differential predictors between the first and second trimester models are thyroid. Therefore, the comportment of the models' performance is in agreement with what was already discussed: thyroid markers are useful for the prediction of GDM, especially at the second trimester of pregnancy.

In regression, the full model with first trimester data alone got a RECV of 21.4%. The selection of the 7 most relevant variables decreased the RECV to 20.7%. In the case of regression comprising second trimester data, the full model yielded a RECV of 18.7%. The selection of the 9 most important variables improved the predictive performance, but in a more discreet degree than in the first trimester case (RECV of 18.4%). Even though none of the simplified

models include first trimester thyroid predictors, the latter does contain second trimester thyroid markers. Similar to what was observed in classification, the second trimester model has a better predictive performance than the first trimester model, both before and after variable selection. Together, these observations reaffirm the idea that second trimester thyroid variables are useful for the prediction of OGTT post load glycemia.

Fig 4 presents the cross-validation predictive performance of the final simplified models in a graphical manner. Fig 4A and 4B display the receiver operating characteristic (ROC) curve for the prediction of GDM in the first and the second trimester of pregnancy, respectively. Fig 4C and 4D show the correlation between the actual and the predicted value of post load glycemia in the first and second trimester of pregnancy, respectively. Calibration plots are exhibited in S1 Fig.

The prediction of GDM with the first trimester model reaches calibration and cross-validation AUCs of 0.914 (95% CI: 0.821–1.000, p<0.001****) and 0.867 (95% CI: 0.741–0.994, p<0.001****), respectively. This classification model covers the top 4 most important predictors presented in Fig 2A, i.e. personal history of anemia, FTO genotype, prior GDM and family history of DM2. The contribution of each predictor to the final model is shown on Panel A in S2 Fig. The relationship between them and GDM was already discussed. Remarkably, the deletion of FTO genotype from the model decreases calibration and cross-validation AUCs to 0.860 (95% CI: 0.722–0.998, p<0.001****) and 0.759 (95% CI: 0.556–0.962, p<0.01**), respectively.

GDM prediction with the second trimester model achieves AUCs of 0.940 (95% CI: 0.860–1.000, p<0.001****) and 0.920 (95% CI: 0.823–1.000, p<0.001****) in calibration and cross-validation, respectively. This classification model comprises the top 9 most relevant variables displayed in Fig 2B, i.e. TSH2T, prior GDM, family history of DM2, FT42T, personal history of anemia, TG2T, TG1T, BMI and past HELLP syndrome. The contribution of each predictor to the final model is presented on Panel B in S2 Fig. The association between the majority of them and GDM was formerly discussed, except for past hemolysis, elevated liver enzymes and low platelet (HELLP) syndrome. Past HELLP syndrome tends to be more frequent in GDM than in NGT pregnancies (S1 Table). The relationship between this syndrome and the development of GDM in a future pregnancy has not been studied. However, HELLP has been linked to long-term glucose intolerance [56]. Furthermore, severe hypertensive pregnancy disorders, including the HELLP syndrome, have been associated with an increased risk of DM2 [57]. Interestingly, the remotion of all biochemical variables from the model, i.e. TSH2T, FT42T, TG2T and TG1T, reduces the predictive accuracy in both calibration and cross-validation, with AUCs of 0.914 (95% CI: 0.836–0.992, p<0.001****) and 0.819 (95% CI: 0.682–0.957, p<0.001***), respectively. This result reinforces the idea that thyroid markers are useful to predict GDM at the second trimester of pregnancy.

It is important to note that the classification models presented in this study were calibrated (i.e. developed or trained), optimized and then validated using the Chilean diagnostic criteria. Therefore, they should only be used to predict GDM under that criteria. Very few articles have proposed strategies to predict GDM in Chilean pregnant women. They used clinical parameters [23], plasma biomarkers [58], plasma extracellular vesicles [59], oral extracellular vesicles [60] and the combination of glycemia with BMI or gingival crevicular fluid placental growth factor (GCF-PlGF) [61]; and got AUCs of 0.739, 0.870, 0.798, 0.814, 0.828 and 0.898, respectively. Those studies employed preconception [23] or first trimester data [58–61], and the only ML technique used was LR. None of them validated their models, hence, the reported AUCs are calibration AUCs. The first and second trimester models presented here have similar or better calibration AUCs. Moreover, they are cross-validated. As far as we know, this is the first study that develops and validates ML models for the prediction of GDM in Chilean subjects.

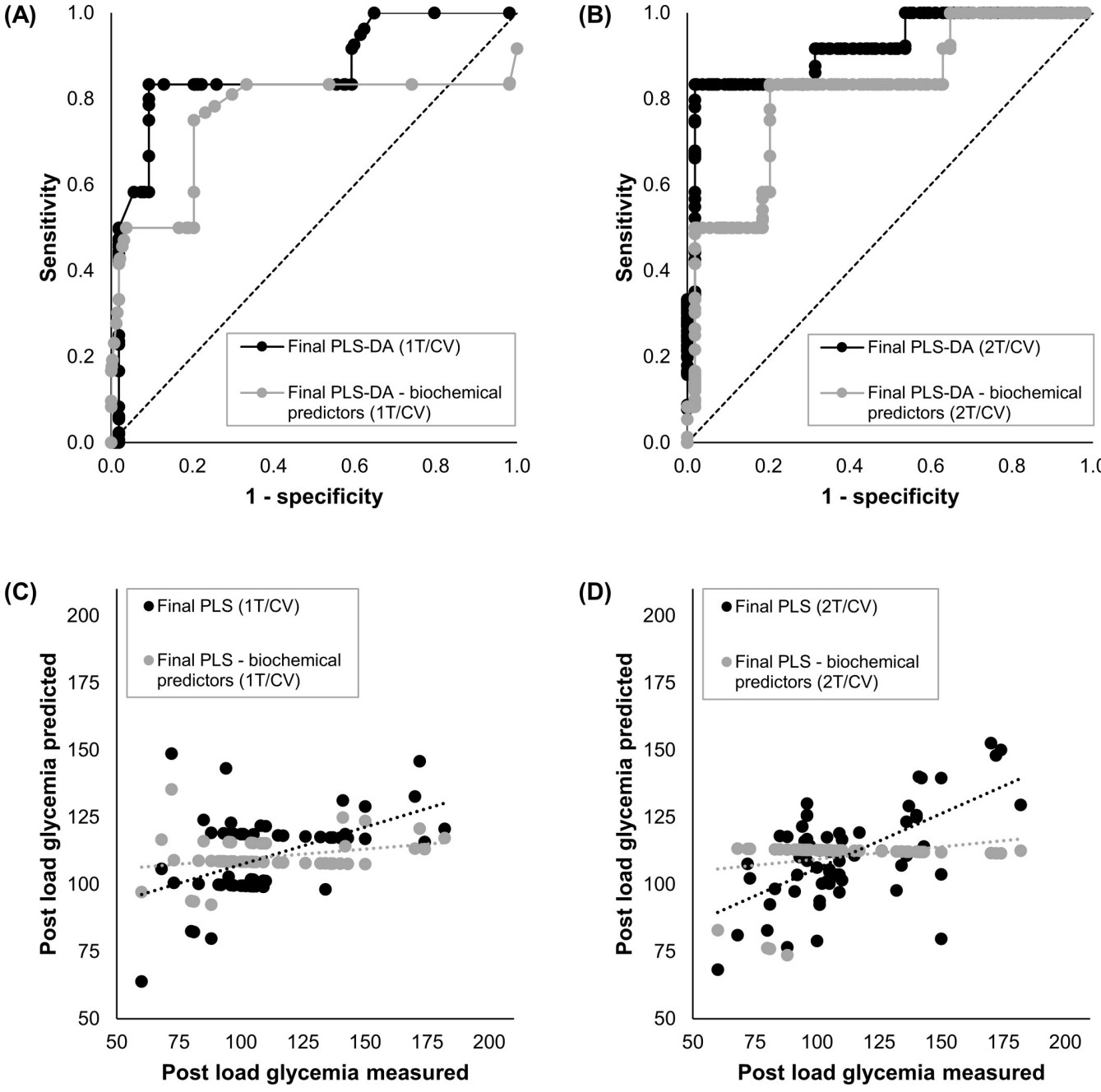

**Fig 4. Cross-validation performance of the final machine learning models that predict gestational diabetes mellitus and post load glycemia.** A-B) For the prediction of GDM in the first (A) or second (B) trimester of pregnancy by PLS-DA. C-D) For the prediction of post load glycemia in the first (C) or second (D) trimester of pregnancy by PLS. The four plots show the cross-validation performance of the final models with their full set of predictors (black) and after the remotion of biochemical markers (grey). *1T*: *First trimester. 2T*: *Second trimester. PLS-DA*: *Partial least squares discriminant analysis. PLS*: *Partial least squares. CV*: *Cross-validation.*

Post load glycemia prediction with the first trimester model yields a Spearman r correlation coefficient of 0.496 (95% CI: 0.282–0.663, p<0.001\*\*\*\*) and 0.259 (95% CI: 0.011–0.477, p = 0.036\*) in calibration and cross-validation, respectively. This regression model encompasses the top 7 most important predictors exhibited in Fig 3A, i.e. FTO genotype, past HELLP

syndrome, prior septicemia, prior pleural effusion, past asthma, family history of DM2 and prior GDM. The contribution of each predictor to the final model is presented on Panel A in S3 Fig. The link between most of them and GDM was discussed above, except for prior septicemia, prior pleural effusion and past asthma. Prior septicemia tends to be more frequent in GDM than in NGT pregnancies (S1 Table). Even though its association with GDM has not been investigated, septicemia has been related to a higher risk of DM2 [62,63]. Likewise, prior pleural effusion leans to be more frequent in GDM than in NGT subjects (S1 Table). There are several long-term follow-up studies of patients affected with pleural effusion [64–69], but none of them assessed diabetes or related conditions as outcomes. Hence, more studies are needed to explain the link between pleural effusion and GDM. At last, past asthma tends to be less common in GDM than in NGT pregnant women (S1 Table). In contrast, a meta-analysis reported that maternal asthma was associated with a higher risk of GDM [70]; however, it didn't evaluate the risk of GDM by ethnicity. A different study revealed that Hispanic subjects with and without asthma had an increased risk of GDM, when compared to White subjects. Interestingly, in the Hispanic cohort, the non-asthmatics had a higher odds ratio (OR) for GDM than the asthmatics [71]. It is interesting to note that, as in the classification first trimester final model, the deletion of FTO genotype from the regression model is associated with an impaired predictive performance, with calibration and cross-validation Spearman r correlation coefficients of 0.375 (95% CI: 0.139–0.571, $p < 0.01^{**}$) and -0.081 (95% CI: -0.324–0.171, $p > 0.05$ ns), respectively.

The prediction of post load glycemia with the second trimester model gets calibration and cross-validation Spearman r correlation coefficients of 0.520 (95% CI: 0.311–0.681, $p < 0.001^{****}$) and 0.457 (95% CI: 0.234–0.634, $p < 0.001^{***}$), respectively. This regression model involves the top 9 most relevant variables shown in Fig 3B, i.e. TSH2T, FT42T, FBG2T, TT42T, FTO genotype, past HELLP syndrome, prior septicemia, prior pleural effusion and past asthma. The contribution of each predictor to the final model is exhibited on Panel B in S3 Fig. Their relation with GDM was already discussed. Remarkably, as in the classification second trimester final model, the remotion of all biochemical variables from the regression model, i.e. TSH2T, FT42T, FBG2T, TT42T and FTO genotype, leads to a reduced predictive performance, with Spearman r correlation coefficients of 0.425 (95% CI: 0.197–0.609, $p < 0.001^{***}$) and -0.679 (95% CI: -0.794–0.518, $p < 0.001^{****}$) in calibration and cross validation, respectively. This result supports the idea that thyroid-related markers are useful to predict post load glycemia at the second trimester of pregnancy.

It is important to note that, contrary to what happens in the case of classification models, the regression models presented in this work are not restricted to a specific diagnostic criteria. This is because they allow to predict the post load glycemia very value, and then that value can be interpreted as NGT or GDM depending on the criteria applied. The error associated with the prediction of post load glycemia (given in terms of RECV) using the best regression models of this study is 20.7% in the first trimester of pregnancy and 18.4% in the second trimester of pregnancy, regardless of the diagnostic criteria employed. Worldwide, few studies have proposed strategies to predict glycemia in the context of pregnancy. Most of them are focused on blood glucose monitoring for the management of GDM [72–74], and not on the prediction of post load glycemia as an early detection or alternative screening tool for this pregnancy disorder. Furthermore, although some of them evaluate the correlation between different variables and fasting glycemia [75–77], postprandial glycemia [78], glucose challenge test (GCT) post load glycemia [79,80] and OGTT post load glycemia [81–83], they don't test their predictive performance through error metrics like RMSE and RE. To our knowledge, this is the first study that introduces and properly assesses ML models to predict OGTT post load glycemia so as to make the early detection and screening of GDM independent of the diagnostic criteria.

### 3.6. Strengths and limitations of this study

This study has some limitations. The sample size is small, thus, the generalization of the findings described here should be made with moderation. Models built on a restricted number of subjects are prone to overfitting. Even though internal validation was carried out to reduce the risk of it, the possibility that the presented models may be associated with some degree of overfitting cannot be ruled out. Future external validation studies are needed to evaluate that possibility. In addition, some of the clinical variables used were self-reported. Hence, the association between those particular parameters and GDM should be interpreted with caution.

Despite its limitations, this investigation has several strengths. Differently from previous studies, it focuses on the capability of the whole thyroid profile to predict GDM, both separately and in association with other maternal variables. Moreover, this work considered thyroid markers from the first and the second trimester of pregnancy, i.e. it tested their potential for both the early detection and the alternative screening of GDM. In addition, it assessed the power of thyroid and other maternal parameters on the prediction of OGTT post load glycemia, which hadn't been done before. All the presented models were cross-validated, which reduces the chance of overestimating their predictive performance. To our knowledge, this is the first study that performs such a systematic analysis to evaluate the effectiveness of thyroid variables for the prediction of GDM and OGTT post load glycemia.

## 4. Conclusion

In this work, the maternal thyroid profile from the first and the second trimester of pregnancy was methodically assessed for the prediction of GDM and OGTT post load glycemia, both alone and combined with other variables. Our results indicate that thyroid parameters are useful for the prediction of GDM and OGTT post load glycemia, especially at the second trimester of pregnancy. Therefore, these markers could be employed as an alternative screening tool for GDM, independently of the diagnostic criteria used. Further studies are required to confirm these findings in larger cohorts and different populations.

Despite its limitations, this investigation represents the first attempt to evaluate the effectiveness of the whole thyroid profile on the prediction of GDM and OGTT post load glycemia. Our best models achieved a good predictive performance for both outcomes. Future external validation studies are needed so as to corroborate their potential and bring them closer to clinical practice.

## Supporting information

**S1 Fig. Calibration performance of the final machine learning models that predict gestational diabetes mellitus and post load glycemia.** A-B) For the prediction of GDM in the first (A) or second (B) trimester of pregnancy by PLS-DA. C-D) For the prediction of post load glycemia in the first (C) or second (D) trimester of pregnancy by PLS. The four plots show the calibration performance of the final models with their full set of predictors (black) and after the remotion of biochemical markers (grey). *1T*: *First trimester*. *2T*: *Second trimester*. *PLS-DA*: *Partial least squares discriminant analysis*. *PLS*: *Partial least squares*. *CV*: *Cross-validation*. (TIF)

**S2 Fig. Variable importance of the machine learning models that predict gestational diabetes mellitus with best performance after variable selection.** (A) Built on first trimester data only. (B) Including second trimester data. *1T*: *First trimester*. *2T*: *Second trimester*. *PLS-DA*: *Partial least squares discriminant analysis*. *pGDM*: *Prior gestational diabetes mellitus*. *fDM2*: *Family history of type 2 diabetes*. *mAnemia*: *Personal anemia*. *FTO*: *Fat mass and obesity-*

*associated genotype (rs9939609). TSH: Thyroid stimulating hormone. FT4: Free thyroxine. TG: Thyroglobulin. BMI: Body mass index. pHELLP: Prior hemolysis elevated liver enzymes and low platelets syndrome.*
(TIF)

**S3 Fig. Variable importance of the machine learning models that predict post load glycemia with best performance after variable selection.** (A) Built on first trimester data only. (B) Including second trimester data. *1T: First trimester. 2T: Second trimester. PLS: Partial least squares. FTO: Fat mass and obesity-associated genotype (rs9939609). mAsthma: Personal asthma. pGDM: Prior gestational diabetes mellitus. pHELLP: Prior hemolysis elevated liver enzymes and low platelets syndrome. pSepticemia: Prior septicemia. pPEffusion: Prior pleural effusion. fDM2: Family history of type 2 diabetes. TSH: Thyroid stimulating hormone. FBG: Fasting blood glucose. FT4: Free thyroxine. TT4: Total thyroxine.*
(TIF)

**S1 Table. Maternal variables that were recorded in both normal glucose tolerance and gestational diabetes mellitus pregnancies.** NGT: Normal glucose tolerance. GDM: Gestational diabetes mellitus. BMI: Body mass index. HELLP: Hemolysis elevated liver enzymes and low platelets syndrome. IUGR: Intrauterine growth restriction. PCOS: Polycystic ovary syndrome. DM1: Type 1 diabetes. DM2: Type 2 diabetes. FTO: Fat mass and obesity-associated gene. TSH: Thyroid stimulating hormone. TT3: Total triiodothyronine. TT4: Total thyroxine. FT4: Free T4. TG: Thyroglobulin. aTG: TG antibody. aTPO thyroid peroxidase antibody. TRAb: TSH receptor antibody. OGTT: Oral glucose tolerance test. Qualitative variables are presented as percentage (proportion); quantitative variables with normal distribution, as mean ± standard deviation; and quantitative variables with non-normal distribution, as median (interquartile range). NS: Not significant. * p<0.05. ** p<0.01. **** p<0.0001.
(DOCX)

**S2 Table. Maternal thyroid variables that differed significantly between normal glucose tolerance and gestational diabetes mellitus pregnancies.** TSH: Thyroid stimulating hormone. TT3: Total triiodothyronine. FT4: Free thyroxine. aTG: Thyroglobulin antibody. Quantitative variables with non-normal distribution are presented as median (interquartile range). NS: Not significant. * p<0.05. ** p<0.01. *** p<0.001. **** p<0.0001.
(DOCX)

**S3 Table. Sensitivity and specificity of machine learning models that predict gestational diabetes mellitus based on different maternal variables.** [a] Model with the highest cross-validation non-error rate (CV-NER) for GDM prediction, using 1T data only. [b] Model with the highest CV-NER for GDM prediction, including 2T data. LR: Logistic regression. L-SVM: Linear support vector machine. PLS-DA: Partial least squares discriminant analysis. CART: Classification and regression tree. XGB: Extreme gradient boosting. 1T: First trimester. 2T: Second trimester. Thy: Thyroid predictors. NoThy: Non-thyroid predictors.
(DOCX)

**S4 Table. Root mean square error of machine learning models that predict post load glycemia based on different maternal variables.** [a] Model with the lowers RE for GDM prediction, using 1T data only. [b] Model with the lowest RE for GDM prediction, including 2T data. PLS: Partial least squares. RMSEC: Root mean square error of calibration. RMSECV: Root mean square error of cross-validation. 1T: First trimester. 2T: Second trimester. Thy: Thyroid predictors. NoThy: Non-thyroid predictors.
(DOCX)

**S5 Table. Sensitivity, specificity, non-error rate, root mean square error and relative error of machine learning models that predict gestational diabetes mellitus and post load glycemia before and after variable selection.** [a] For PLS-DA: NoThy1T + Thy1T. For PLS: NoThy1T. [b] Model with the lowest RE for post load glycemia prediction, using 1T data only. Predictors: FTO, pHELLP, pSepticemia, pPEffusion, mAsthma, fDM2, pGDM. [c] Model with the highest NER for GDM prediction, using 1T data only. Predictors: mAnemia, FTO, pGDM, fDM2. [d] For PLS-DA: NoThy1T + Thy1T + Thy2T. For PLS: NoThy1T + NoThy2T + Thy2T. [e] Model with the highest NER for GDM prediction, including 2T data. Predictors: TSH2T, pGDM, fDM2, FT42T, mAnemia, TG2T, TG1T, BMI, pHELLP. [f] Model with the lowest RE for post load glycemia prediction, including 2T data. Predictors: TSH2T, FT42T, FBG2T, TT42T, FTO, pHELLP, pSepticemia, pPEffusion, mAsthma. PLS-DA: Partial least squares discriminant analysis. PLS: Partial least squares. Se: Sensitivity. Sp: Specificity. NER: Non-error rate. RMSE: Root mean square error. RE: Relative error. NoThy: Non-thyroid predictors. Thy: Thyroid predictors. 1T: First trimester. 2T: Second trimester. FTO: Fat mass and obesity-associated genotype (rs9939609). pHELLP: Prior hemolysis elevated liver enzymes and low platelets syndrome. pSepticemia: Prior septicemia. pPEffusion: Prior pleural effusion. mAsthma: Personal asthma. fDM2: Family history of type 2 diabetes. pGDM: Prior gestational diabetes mellitus. mAnemia: Personal anemia. TSH: Thyroid stimulating hormone. FT4: Free thyroxine. TG: Thyroglobulin. BMI: Body mass index. FBG: Fasting blood glucose. TT4: Total thyroxine.
(DOCX)

**S1 Dataset. Full data analyzed in this study.** FTO: Fat mass and obesity-associated. BMI: Body mass index. TSH: Thyroid stimulating hormone. T3: Triiodothyronine. T4: Thyroxine. TG: Thyroglobulin. aTG: TG antibody. aTPO: Thyroid peroxidase antibody. TRAb: TSH receptor antibody. GDM: Gestational diabetes mellitus. HELLP: Hemolysis elevated liver enzymes and low platelets. IUGR: Intrauterine growth restriction. PCOS: Polycystic ovary syndrome. DM1: Type 1 diabetes. DM2: Type 2 diabetes. OGTT: Oral glucose tolerance test. 1T: First trimester. 2T: Second trimester.
(XLSX)

## Acknowledgments

The authors would like to thank all the pregnant women who voluntarily participated in this study; and to the health staff of the primary health centers (CESFAM) Victor Manuel Fernández, Santa Sabina and Tucapel, who collaborated in the collection and the initial storage of blood samples, and the retrieval of institution-derived maternal data.

## Author Contributions

**Conceptualization:** Daniela Mennickent, Bernel Ortega-Contreras, Juan Araya, Enrique Guzmán-Gutiérrez.

**Data curation:** Daniela Mennickent, Bernel Ortega-Contreras, Sebastián Gutiérrez-Vega, Enrique Guzmán-Gutiérrez.

**Formal analysis:** Daniela Mennickent, Juan Araya.

**Funding acquisition:** Daniela Mennickent, Bernel Ortega-Contreras, Andrés Rodríguez, Juan Araya, Enrique Guzmán-Gutiérrez.

**Investigation:** Daniela Mennickent, Bernel Ortega-Contreras, Sebastián Gutiérrez-Vega, Erica Castro.

**Methodology:** Daniela Mennickent, Juan Araya, Enrique Guzmán-Gutiérrez.

**Project administration:** Juan Araya, Enrique Guzmán-Gutiérrez.

**Resources:** Erica Castro, Andrés Rodríguez, Juan Araya, Enrique Guzmán-Gutiérrez.

**Software:** Daniela Mennickent, Juan Araya.

**Supervision:** Juan Araya, Enrique Guzmán-Gutiérrez.

**Validation:** Daniela Mennickent, Juan Araya.

**Visualization:** Daniela Mennickent, Andrés Rodríguez, Juan Araya, Enrique Guzmán-Gutiérrez.

**Writing – original draft:** Daniela Mennickent, Enrique Guzmán-Gutiérrez.

**Writing – review & editing:** Bernel Ortega-Contreras, Sebastián Gutiérrez-Vega, Erica Castro, Andrés Rodríguez, Juan Araya, Enrique Guzmán-Gutiérrez.

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
