## [Decision Letter · Decision Letter 0]

25 Oct 2022

PONE-D-21-39287Evaluation of first and second trimester maternal thyroid profile on the prediction of gestational diabetes mellitus and post load glycemia.PLOS ONE

Dear Dr. Enrique Guzman,

Thank you for submitting your manuscript to PLOS ONE. After careful consideration, we feel that it has merit but does not fully meet PLOS ONE’s publication criteria as it currently stands. Therefore, we invite you to submit a revised version of the manuscript that addresses the points raised during the review process.

We look forward to receiving your revised manuscript.

Kind regards,

Surangi Nilanka Jayakody Mudiyanselage, MBBS, MD

Academic Editor

PLOS ONE

“This work was financially supported by ANID (https://www.anid.cl/) through FONDECYT 11170710 (EG-G), FONDECYT 11181153 (JA), PhD scholarship 21190736 (DM) and MSc scholarship 22201750 (BO-C); Universidad de Concepción/MINEDUC (https://www.udec.cl/ and https://www.mineduc.cl/) via Project UCO 1866 fund (DM); and Universidad del Bio-Bío (https://www.ubiobio.cl/) by means of 180709 3/R DIUBB grant (AR). The funders had no role in study design, data collection and analysis, decision to publish, or preparation of the manuscript.”

Reviewers' comments:

Reviewer's Responses to Questions

**Comments to the Author**

1. Is the manuscript technically sound, and do the data support the conclusions?

Reviewer #1: Partly

Reviewer #2: Partly

2. Has the statistical analysis been performed appropriately and rigorously? 

Reviewer #1: 

Reviewer #2: Yes

3. Have the authors made all data underlying the findings in their manuscript fully available?

Reviewer #1: Yes

Reviewer #2: Yes

4. Is the manuscript presented in an intelligible fashion and written in standard English?

Reviewer #1: Yes

Reviewer #2: Yes

5. Review Comments to the Author

Reviewer #1:

This is very interesting study with wonderful finding , however, small sample size is my major concern. 

Reviewer #2: Review JCM-1633354

The paper by Mennickent et al. describes the maternal thyroid profile in first and second trimester and its role on the prediction of GDM and post load glycemia. The study is a small case-control study with 66 participants with blood sampling in first trimester and second trimester (at GDM diagnostic OGTT).

The authors perform advanced multivariate analysis to come up with predictive models.

Major points:

1) As the authors point out themselves, the sample size is too small and there is a major concern of overfitting despite the use of leave-one-out cross-validation.

2) The data in the paper is not supporting the title, since Table 1 does not present First trimester thyroid profiles.

3) The term prediction should then be changed to association, since the study really looks at the association between 2nd trimester thyroid profiles and GDM/OGTT.

4) There is a major use of risk factors in the models (non-thyroid variables) and these are in the introduction stated as poor performing, yet for example prior GDM is the third most important variable and these risk factors are widely used in the models.

Also for risk factors, important literature in favor of risk factors is missing i.e. PMID: 31932807 (page 3, line 53)

5) Please report how the different variables are contributing to the final classification models and please state and make it clear to the reader which variables are in the final models.

6) GDM diagnostic criteria; How are the performance of the models with WHO criteria? At least a comparison to the 2-h 8.5 mM threshold is important.

Minor points:

7) How was fasting glucose evaluated in this study?

8) TSH, TT3 and aTG were higher in the GDM group but could these differences be ascribed to differences in gestation age? Please include this information in Table 1. And discuss how these markers change in the second trimester.

9) Please do not use abbreviations in table headings.

10) Tables 2+3, could be supplementary.

11) Figure 4: It would be much more helpful to the reader to see ROC curves of core risk factors alone and + biochemical/genetic variables for both first and second trimester.

6. PLOS authors have the option to publish the peer review history of their article (what does this mean?). If published, this will include your full peer review and any attached files.

Reviewer #1: No

Reviewer #2: No

---

## [Author Response · Author response to Decision Letter 0]

5 Dec 2022

RESPONSE TO REVIEWERS

Academic Editor

R: We checked PLOS ONE style templates and submission guidelines, and made the necessary changes to bring our manuscript into compliance with the journal requirements.

“This work was financially supported by ANID (https://www.anid.cl/) through FONDECYT 11170710 (EG-G), FONDECYT 11181153 (JA), PhD scholarship 21190736 (DM) and MSc scholarship 22201750 (BO-C); Universidad de Concepción/MINEDUC (https://www.udec.cl/ and https://www.mineduc.cl/) via Project UCO 1866 fund (DM); and Universidad del Bio-Bío (https://www.ubiobio.cl/) by means of 180709 3/R DIUBB grant (AR). The funders had no role in study design, data collection and analysis, decision to publish, or preparation of the manuscript. ”

R: We included the amended Funding Statement in our Cover Letter.

Reviewer #1

1. This is very interesting study with wonderful finding , however, small sample size is my major concern. 

R: Thank you for your comment. We recognize that sample size is a limitation of our work, as stated in Section 3.6 “Strengths and limitations of this study”. Most of the studies that aim to predict GDM in Chilean subjects have similar sample sizes, with n=28 and n=14 for control and GDM, respectively (https://doi.org/10.1002/jper.17-0497); n=80 and n=16 for control and GDM, respectively (https://doi.org/10.1515/jpm-2018-0120); n=19 and n=6 for control and GDM, respectively (https://doi.org/10.4067/S0034-98872019001201503); and n=23 and n=11 for control and GDM, respectively (https://doi.org/10.1371/journal.pone.0218616). Notably, the last one is published in PLOS ONE. None of the mentioned articles validated their models. We did. To our knowledge, ours is the first study that actually validates a predictive model for GDM in Chilean pregnant women. Therefore, we think that our results are worthy to be shared with the scientific community, despite small sample size. In addition, as far as we know there are no published articles that aim to predict the OGTT post load glycemia as a way to make the early detection and screening of GDM independent of the diagnostic criteria. Ours does. In spite of small sample size, we strongly believe that this study is a very important contribution to the field, as it not only presents, but is also a proof of concept for this novel approach. Finally, but not less important, there are no papers in literature that systematically assess the effectiveness of first and second trimester complete maternal thyroid profiles on the prediction of GDM and OGTT post load glycemia. Our does. Despite its limitations, we consider that our study represents a first step to understand the potential that thyroid markers have to predict GDM, independent of the diagnostic criteria used.

Reviewer #2

The paper by Mennickent et al. describes the maternal thyroid profile in first and second trimester and its role on the prediction of GDM and post load glycemia. The study is a small case-control study with 66 participants with blood sampling in first trimester and second trimester (at GDM diagnostic OGTT). The authors perform advanced multivariate analysis to come up with predictive models.

Major points:

1) As the authors point out themselves, the sample size is too small and there is a major concern of overfitting despite the use of leave-one-out cross-validation.

R: Thank you for your comment. We are aware that the sample size is small, and that this is a limitation of our study. Models built on a small number of samples are prone to overfitting, and the way to reduce the risk of it is through internal validation, such as leave-one-out cross-validation. Strictly speaking, cross-validation doesn’t avoid overfitting but lowers the risk of it occurring. Taking that into consideration, any model built on the basis of a small number of samples, although internally validated, is associated with a certain risk of overfitting. The manner to evaluate that is through external validation. We modified the text in order to account for this limitation more explicitly (lines 29, 41, 43-44, 611-614, 623, 636, 639).

We would like to add that most of the studies that aim to predict GDM in Chilean subjects have sample sizes similar to ours, with n=28 and n=14 for control and GDM, respectively (https://doi.org/10.1002/jper.17-0497); n=80 and n=16 for control and GDM, respectively (https://doi.org/10.1515/jpm-2018-0120); n=19 and n=6 for control and GDM, respectively (https://doi.org/10.4067/S0034-98872019001201503); and n=23 and n=11 for control and GDM, respectively (https://doi.org/10.1371/journal.pone.0218616). Notably, the last one is published in PLOS ONE. None of the mentioned studies performed validation procedures to reduce the risk of overfitting. We did. To our knowledge, ours is the first study that actually validates a predictive model for GDM in Chilean pregnant women. Therefore, we think that our results are worthy to be shared with the scientific community, despite small sample size. Moreover, as far as we know there are no published articles that aim to predict the OGTT post load glycemia as a way to make the early detection and screening of GDM independent of the diagnostic criteria. Ours does. In spite of small sample size, we strongly believe that this study is a very important contribution to the field, as it not only presents, but is also a proof of concept for this novel approach. Finally, but not less important, there are no papers in literature that systematically assess the effectiveness of first and second trimester complete maternal thyroid profiles on the prediction of GDM and OGTT post load glycemia. Our does. Despite its limitations, we consider that our study represents a first step to understand the potential that thyroid markers have to predict GDM, independent of the diagnostic criteria used.

2) The data in the paper is not supporting the title, since Table 1 does not present First trimester thyroid profiles.

R: We appreciate this comment. In this work we used a total of 75 maternal variables, encompassing first and second trimester thyroid profiles. Table 1 shows only the variables that emerged as statistically significant in univariate analyses. The remaining variables, which include all the first trimester thyroid markers, are presented in S1 Table. We modified the text in order to make this clearer (lines 201-203).

3) The term prediction should then be changed to association, since the study really looks at the association between 2nd trimester thyroid profiles and GDM/OGTT.

R: Thank you for your comment. As we clarified in question 2, this study evaluates both first and second trimester thyroid profiles on the prediction of GDM and post load glycemia. We understand that “prediction” could be a tricky word when we are using second trimester markers to get to know outcomes that also occur at the second trimester. However, in the field of machine learning, the term “prediction” refers to forecast a future behavior or an unobserved outcome (definition from https://doi.org/10.1038/nmeth.4642). The former part of the definition is valid for first trimester variables, which allow to “predict” (as an early detection tool) GDM and post load glycemia; and the later part of the definition is valid for second trimester variables, which allow to “predict” (as an alternative screening tool) GDM and post load glycemia. Then, the term “prediction” is valid for both the first and the second trimester thyroid profiles. We included the above definition in Section 1 “Introduction” (lines 50-52).

4) There is a major use of risk factors in the models (non-thyroid variables) and these are in the introduction stated as poor performing, yet for example prior GDM is the third most important variable and these risk factors are widely used in the models.

Also for risk factors, important literature in favor of risk factors is missing i.e. PMID: 31932807 (page 3, line 53)

R: We agree: The original text was imprecise. Literature shows that models that are built only on classic clinical risk factors are typically associated with a poor predictive performance. However, the addition of biochemical risk factors generally leads to an improved accuracy. The reference you suggested (PMID: 31932807) shows exactly that: Both their full model (including more than 2000 predictors) and their simplified model (consisting of nine questions) combined clinical and biochemical risk factors, and both outperformed a baseline model built only on classic clinical risk factors that had a poor predictive performance (AUC = 0.68). It is in that context in which it is necessary to evaluate other biomarkers. We modified the text and included new references in order to present this idea in a clearer way (lines 53, 55-57). Thank you.

5) Please report how the different variables are contributing to the final classification models and please state and make it clear to the reader which variables are in the final models.

R: We modified the text in order to explicitly state the variables that are included in each final model, and added two supplementary figures (S2 and S3 Figs) so as to report their contribution to them (lines 525-526, 533-535, 561-563 and 583-585). Thank you for the suggestion.

6) GDM diagnostic criteria; How are the performance of the models with WHO criteria? At least a comparison to the 2-h 8.5 mM threshold is important.

R: Thank you for this comment. Regarding classification models: They were calibrated (i.e. developed or trained), optimized and then validated using the Chilean criteria, therefore they are restricted to that criteria. In other words, it wouldn’t be correct to test them with a different diagnostic criteria. Regarding regression models: They were built to overcome the restriction that classification models have, as they allow to predict the post load glycemia very value, and then that value can be interpreted as non-GDM or GDM employing any diagnostic criteria. Regardless of the criteria used, the error associated with the prediction of post load glycemia is 20.7% applying the best first trimester regression model, and 18.4% applying the best second trimester regression model (both errors from cross-validation analyses). We modified the text in order to make the above clearer (lines 546-548 and 592-598). 

Minor points:

7) How was fasting glucose evaluated in this study?

R: Fasting glucose was measured in sodium fluoride/citrate plasma from pregnant women with 12 hours of fasting, using the hexokinase method. This information was not in the original manuscript, we included it in Section 2 “Materials and methods” (lines 88-90, 110). Thank you. 

8) TSH, TT3 and aTG were higher in the GDM group but could these differences be ascribed to differences in gestation age? Please include this information in Table 1. And discuss how these markers change in the second trimester.

R: Thank you for your comment. Regarding gestational age: We only have the exact gestational age for a subset of subjects from the full cohort. Using that partial information and classic univariate techniques we compared the gestational age of NGT and GDM groups at the first and second blood sampling, and there were no statistically significant differences between them (for the first blood sampling: n=41 for NGT, n=9 for GDM, p=0.994; for the second blood sampling: n=30 for NGT, n=8 for GDM, p= 0.744). Considering that all patients were recruited in the same manner, and that only at the end we knew if they had NGT or GDM, we would expect the same behavior for the full cohort. Hence, it is unlikely that the differences observed between the two groups are due to differences in gestational age. Nevertheless, since we do not have the exact gestational age of all the patients involved in this study, and the measures of central tendency and dispersion obtained are not exactly the same as those that would be obtained for the full cohort, we think that it wouldn’t be entirely correct to include this information in Table 1. Regarding how the mentioned markers change in the second trimester of pregnancy: We added it to the text (lines 241-268). 

9) Please do not use abbreviations in table headings.

R: We replaced abbreviations for full words in table and figure titles. Thank you.

10) Tables 2+3, could be supplementary.

R: Even though we carefully considered this suggestion, we decided to keep Tables 2 and 3 in the main text. The text that is immediately before Tables 2 and 3 introduces concepts that are explained for the first time in the manuscript. We consider that these tables help to understand those concepts. Moreover, the text that is immediately after Tables 2 and 3 explicitly refers to results that are displayed in them. We think that it’s easier to contrast the written information with the tabulated information if both are within the same document.

11) Figure 4: It would be much more helpful to the reader to see ROC curves of core risk factors alone and + biochemical/genetic variables for both first and second trimester.

R: Fig 4 and S1 Fig were modified in order to include what you suggested. The text was also modified so as to explain it (lines 527-529, 541-545, 576-579, 586-591). Thank you.

---

## [Editor Report · Decision Letter 1]

3 Jan 2023

Evaluation of first and second trimester maternal thyroid profile on the prediction of gestational diabetes mellitus and post load glycemia

PONE-D-21-39287R1

Dear Dr. Enrique,

We’re pleased to inform you that your manuscript has been judged scientifically suitable for publication and will be formally accepted for publication once it meets all outstanding technical requirements.

Kind regards,

Surangi Nilanka Jayakody Mudiyanselage, MBBS, MD

Academic Editor

PLOS ONE

---

## [Editor Report · Acceptance letter]

5 Jan 2023

PONE-D-21-39287R1 

Evaluation of first and second trimester maternal thyroid profile on the prediction of gestational diabetes mellitus and post load glycemia 

Dear Dr. Guzman-Gutierrez:

I'm pleased to inform you that your manuscript has been deemed suitable for publication in PLOS ONE. Congratulations! Your manuscript is now with our production department. 

Kind regards, 

on behalf of

Dr. Surangi Nilanka Jayakody Mudiyanselage 

Academic Editor

PLOS ONE